# Multi-wavelength anomalous diffraction de novo phasing using a two-colour X-ray free-electron laser with wide tunability

Alexander Gorel[1], Koji Motomura[2,3], Hironobu Fukuzawa[2,3], R. Bruce Doak[1], Marie Luise Grünbein[1], Mario Hilpert[1], Ichiro Inoue[3], Marco Kloos[1], Gabriela Kovácsová[1], Eriko Nango[3,4], Karol Nass[1], Christopher M. Roome[1], Robert L. Shoeman[1], Rie Tanaka[3], Kensuke Tono[5], Yasumasa Joti[5], Makina Yabashi [3], So Iwata[3,4], Lutz Foucar[1], Kiyoshi Ueda[2,3], Thomas R.M. Barends[1] & Ilme Schlichting[1]

Serial femtosecond crystallography at X-ray free-electron lasers (XFELs) offers unprecedented possibilities for macromolecular structure determination of systems prone to radiation damage. However, de novo structure determination, i.e., without prior structural knowledge, is complicated by the inherent inaccuracy of serial femtosecond crystallography data. By its very nature, serial femtosecond crystallography data collection entails shot-to-shot fluctuations in X-ray wavelength and intensity as well as variations in crystal size and quality that must be averaged out. Hence, to obtain accurate diffraction intensities for de novo phasing, large numbers of diffraction patterns are required, and, concomitantly large volumes of sample and long X-ray free-electron laser beamtimes. Here we show that serial femtosecond crystallography data collected using simultaneous two-colour X-ray free-electron laser pulses can be used for multiple wavelength anomalous dispersion phasing. The phase angle determination is significantly more accurate than for single-colour phasing. We anticipate that two-colour multiple wavelength anomalous dispersion phasing will enhance structure determination of difficult-to-phase proteins at X-ray free-electron lasers.

[1] Max-Planck-Institut für medizinische Forschung, Jahnstrasse 29, Heidelberg 69120, Germany. [2] Institute of Multidisciplinary Research for Advanced Materials, Tohoku University, Sendai 980-8577, Japan. [3] RIKEN SPring-8 Center, Kouto 1-1-1, Sayo, Hyogo 679-5148, Japan. [4] Department of Cell Biology, Graduate School of Medicine, Kyoto University, Yoshidakonoe-cho, Sakyo-ku, Kyoto 606-8501, Japan. [5] Japan Synchrotron Radiation Research Institute, 1-1-1 Kouto, Sayo-cho, Sayo-gun, Hyogo 679-5198, Japan. Correspondence and requests for materials should be addressed to I.S. (email: Ilme.Schlichting@mpimf-heidelberg.mpg.de)

The bright femtosecond X-ray pulses of X-ray free-electron lasers (XFELs) provide novel opportunities for macromolecular structure determination[1]. In particular, by using a 'diffraction before destruction' approach[2, 3], they allow structure determination of systems prone to radiation damage such as nano- and microcrystals[4–6] or, in many cases, crystals with high-solvent content. The molecules themselves can often be highly radiation sensitive, for example, owing to the presence of metals[7–9] or other redox-sensitive cofactors.

To date, most crystal structures determined via XFEL data collection were solved by molecular replacement using prior structural information for phasing. This approach is suitable when seeking specific information about known protein structures, such as the undamaged active site of a metalloenzyme[7–11] or the nature of a short-lived reaction species as probed in a time-resolved experiment[12–17]. In the long run, however, as XFEL-based data collection matures and also becomes more accessible (with several new XFEL sources coming online this year alone), more and more systems will be studied for which no previous structural information is available. De novo phasing then becomes mandatory. De novo phasing of XFEL data has recently been demonstrated for several model systems, employing a variety of methods based on anomalous signals[18–23] utilising element-specific scattering at X-ray absorption edges or isomorphous differences between native and heavy atom derivatized crystals[5, 24]. Importantly, a previously unknown structure has now also been solved de novo with XFEL data[5].

Despite these successes, de novo phasing of XFEL data remains challenging. This is due to the stochastic nature of XFEL sources and methods of data collection, compounded by current detectors and analysis programmes that limit the accuracy of the integrated diffraction intensities. In contrast to conventional rotation data acquisition, the femtosecond exposure time at XFELs precludes any rotation during exposure and thus results in the collection of still images that contain only partial reflections. Since exposure to the full XFEL beam destroys the illuminated crystal (or at least the illuminated portion thereof), a new crystal, (or a fresh portion), is required for the next exposure. In the case of microcrystals, this must necessarily be a fresh, randomly oriented crystal, leading to a data collection approach termed serial femtosecond crystallography (SFX). The size and quality of microcrystals can vary, however. Moreover, the crystals can intersect the focused XFEL beam anywhere between the low intensity periphery and the high intensity centre of the of X-ray focal spot. Hence, diffraction intensities vary from shot to shot even for identical microcrystals in identical orientations. In addition, the XFEL pulse and photon energy distribution (intensity and wavelength) vary from shot to shot. Together, all of this results in significant fluctuations in the measured intensities that must be averaged out. Consequently, a great deal of data must be collected; the multiplicity of measurements for a given reflection being typically several 100- to 1,000-fold depending on the phasing method and signal strength. This demands not only significant quantities of sample but also of XFEL beam time, both of which are typically precious and often limiting. Improved use of one or both is essential to future evolution of XFEL-based structural biology. To double data collection efficiency, Hunter et al.[22] employed two interaction chambers in series, collecting SFX data using the primary XFEL beam and then 'reusing' the 'spent' XFEL beam after it had passed the first sample and detector[25]. However, this type of data collection does not reduce sample consumption.

The recently established two-colour operation of the SPring-8 Angstrom Compact free-electron LAser (SACLA) in Japan[26] opened up a novel possibility of collecting two SFX datasets simultaneously, without doubling the amount of sample used.

Owing to the unprecedentedly large energy separation of the two tuneable colours of that XFEL beam[26] two distinct and spatially well separated diffraction patterns can be recorded simultaneously on one diffraction image of the same crystal. The simultaneous arrival of the two XFEL pulses precludes damage effects from the first pulse affecting the diffraction of the second pulse[27]. This allows simultaneous same-crystal acquisition of two-wavelength datasets for multiple wavelength anomalous dispersion (MAD) phasing. (This is in marked contrast to data collection at synchrotron sources where they are typically collected sequentially.).

In principle, given the availability of more information, MAD phase angles are expected to be more accurate than those from single wavelength anomalous dispersion (SAD) experiments. To explore whether this can be put to use for XFEL-based de novo phasing with the added benefit of halved sample consumption, we performed a two-colour SFX experiment at SACLA. Using microcrystals of the well-established model system lysozyme, in complex with a lanthanide compound we demonstrate here that simultaneously collected two-colour SFX diffraction data can be analysed and phased de novo. The two colours (7 and 9 keV) were chosen to be above the M-edges and L-edges, respectively (see Fig. 1a). We compare phasing via multiple wavelength and SAD of the two and single-colour SFX data, respectively, and show that the phases are significantly more accurate, facilitating model building for the two-colour data.

## Results

**Experimental set-up and parameter determination.** To test whether two-colour data offer advantages for de novo phasing of SFX data, we used microcrystals of a well-characterised lysozyme heavy atom derivative that gives a strong anomalous signal from two gadolinium atoms per asymmetric unit[28]. This is the same system we employed previously to establish that de novo phasing of SFX data is possible[18]. Lysozyme microcrystals were soaked in gadoteridol, an organic gadolinium complex, and then embedded in a grease matrix[29] for high-viscosity extrusion injection[30] into the XFEL beam at SACLA. Two-colour data collection was performed at beamline 3 (BL3) in the DAPHNIS chamber[31] using a multiport charge coupled device (MPCCD) detector[32] (see Fig. 1b). SACLA operated at 30 Hz and simultaneously delivered two-colour X-ray pulses of 10 fs duration and nominally 7.0 keV ($\lambda = 1.770$ Å) and 9.0 keV ($\lambda = 1.378$ Å) photon energy of 0.14 mJ average power. The focal spot-size was measured to be 1.4 μm (vertical) × 1.6 μm (horizontal) in FWHM and spatial overlap of the two colours was confirmed. To account for the at times higher pulse energy of the 7 keV beam as well as the higher detector quantum efficiency (DQE 0.7 at 7 keV and 0.4 at 9 keV[32] (http://xfel.riken.jp/users/mpccd_detector/instructions_ver1.0_revised.pdf)) and scattering cross sections, we inserted a 25 μm Al filter upstream of the sample that transmitted 60% and 80 % of the 7 keV and 9 keV photons, respectively. We collected 570,000 diffraction patterns in ~12 h. Online data analysis was performed with CASS[33] and the Graphic User Interface to the offline data processing pipeline Cheetah Dispatcher[34] was used to identify 208,373 hits (37% hit rate), using the pipe-line generated geometry file. We used powder patterns of silicon nanocrystals for the accurate determination of the detector distance by applying an interest point algorithm and distance score function optimisation as described in detail in the Supplementary Methods. A wide-range inline spectrometer was used to simultaneously record the spectral information for the 7 keV and 9 keV colours for each XFEL pulse as described in the Methods Section and the Supplementary Note 1. Software modules were implemented to integrate the spectrometer readout into data processing by a

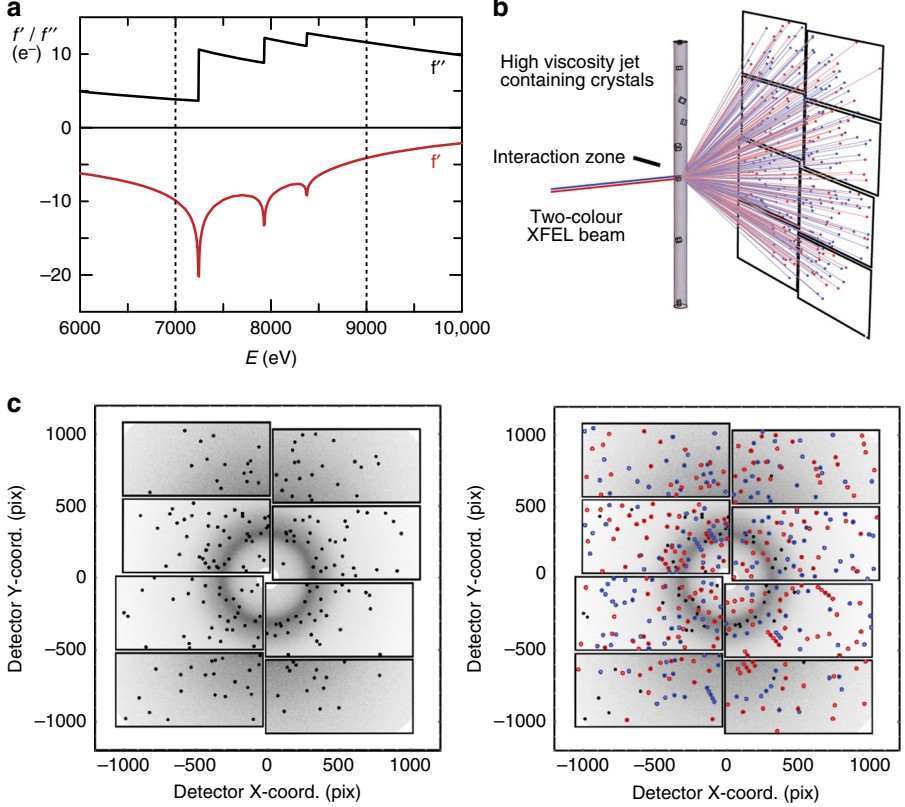

**Fig. 1** Two-colour serial femtosecond crystallography experiment. **a** The two photon energies were chosen to be below (7 keV) and above (9 keV) the L-edges of gadolinium. This results in a strong anomalous difference and a large spatial separation of reflections. **b** Experimental setup. **c** Two-colour diffraction pattern before (*left*) and after (*right*) indexing in the 9 keV colour (*blue*) and 7 keV colour (*red*). The diffuse ring is caused by the grease carrier medium used to deliver the lysozyme crystals into the XFEL beam

**Table 1 Indexing rate of the 208,373 hits at the various stages of the analysis**

| Processing step | Number of indexed images | | |
| --- | --- | --- | --- |
| | 7 keV | 9 keV | 7 and 9 keV |
| No optimisation | 8322 (4%) | 10,374 (5%) | 684 (0.3%) |
| Distance, wavelengths optimised | 15,243 (7.3%) | 23,860 (11.4%) | 2129 (1%) |
| Peaks of dominant pattern removed from search list | 15,243 (7.3%) | 23,860 (11.4%) | 23,144 (11.4%) |

Python interface of the SACLA API (application programming interface) to the metadata database (write_spectra.py) and to add the correct wavelength to the respective diffraction image (write_calib_color.py) so that it can be accessed by the processing software (the indexamajig module from CrystFEL[35]). Supplementary Fig. 1 shows a flowchart of the data analysis. Using the corrected values of the wavelengths and the refined detector distance increased the indexing rate significantly. Out of 208,373 hits we could index 15,243 (7.3 %) in 7 keV, 23,860 (11.4 %) in 9 keV and 2,129 (1%) in both colours (see Fig. 1c, Table 1, Supplementary Figs. 9–11).

**Efficient two-colour data processing**. The two-colour beam is generated in a split undulator operation of the SACLA XFEL. The pulse energies of the two colours can be balanced or adjusted relatively by changing the number of undulators[26]. We aimed at equal distribution, but the pulse energy distribution of the two colours varied during the experiment, with the consequence that the diffraction images typically contained a strong and a weak

diffraction pattern (see Supplementary Fig. 9). This made peak-finding more challenging, given that the analysis software identifies spots in a diffraction pattern by use of intensity thresholding. The strong Bragg reflections from the brighter colour are more likely to lie above the threshold than those of the weaker colour and consequently the list of diffraction spots compiled for indexing will be dominated by spots from the strong pattern. Initial indexing was performed separately for the two colours (see Supplementary Note 2) and yielded the expected unit cell parameters ($a = b = 78.3$ Å, $c = 39.1$ Å, $\alpha = \beta = \gamma = 90°$) for gadoteridol-derivatized lysozyme[18], which were subsequently imposed loosely on indexing. A median filter was applied (see Methods Section for details) to reduce the effects of background and to increase both indexing accuracy and resolution by including weak high resolution reflections into orientation matrix calculation. This resulted in the indexing of the strong diffraction pattern.

To process the second, weaker diffraction pattern in the image, the threshold and minimum $I/\sigma$ values of the peak search parameters were lowered to include weak Bragg reflections (see

**Table 2 SFX data statistics**

| Photon Energy | 7 keV | 9 keV | 9 keV | 7 keV | 9 keV | 7 keV | 9 keV | 7 keV |
|---|---|---|---|---|---|---|---|---|
| Wavelength (Å) | 1.77 | 1.38 | 1.38 | 1.77 | 1.38 | 1.77 | 1.38 | 1.77 |
| Space group | $P4_32_12$ | $P4_32_12$ | $P4_32_12$ | $P4_32_12$ | $P4_32_12$ | $P4_32_12$ | $P4_32_12$ | $P4_32_12$ |
| Unit cell $a = b,c$ (Å) | 78.3,39.1 | 78.3,39.1 | 78.3,39.1 | 78.3,39.1 | 78.3,39.1 | 78.3,39.1 | 78.3,39.1 | 78.3,39.1 |
| No. indexed images[a,b] | 15,234[a] | 23,860[a] | 9,000[b] | 9000[b] | 6000[b] | 6000[b] | 5000[b] | 5000[b] |
| Resolution (Å)[c] | 39.2-2.0 | 35.0-1.9 | 35.0-1.9 | 39.7-2.0 | 35.0-1.9 | 39.7-2.0 | 35.0-1.9 | 39.7-2.0 |
| Completeness (%)[d,c] | 100 (100) | 100 (100) | 100 (100) | 100 (99.7) | 100 (100) | 100 (98.48) | 100 (100) | 100 (96.6) |
| Multiplicity[d,c] | 191 (22) | 214 (151) | 123 (87) | 109 (13) | 81 (57) | 72 (8) | 68 (48) | 61 (7) |
| $R_{split}$[d,e,c] | 17.4 (80) | 13.9 (13.6) | 18.2 (18.3) | 22.5 (100.9) | 22.7 (22.2) | 27.3 (107.3) | 24.8 (24.5) | 29.8(112.0) |
| $CC_{1/2}$[d,c] | 95.2 (34.6) | 95.8 (95.7) | 98.2 (97.8) | 91.6 (16.8) | 97.1 (97.0) | 87.5 (21.7) | 87.6 (86.7) | 85.1 (24.1) |
| $CC_{ano}$[d,c] | [f](12.4) | 23.2 (44.4) | 12.8 (23.3) | [f]($-$[f]) | 5.2 (22.7) | [f](2.7) | 5.1 (23.1) | [f](3.9) |
| $\langle I/\sigma (I)\rangle$[d,c] | 5.3 (1.4) | 7.9 (7.2) | 6.1 (5.6) | 4.17(1.17) | 5.07 (4.71) | 3.51 (1.04) | 4.67 (4.32) | 3.26 (0.9) |

[a]Maximal number of indexed patterns
[b]Subset of randomly selected indexed patterns
[c]The resolution at the edge/corner of the detector was 1.8/1.5 Å (9 keV) and 2.3/1.9 Å (7 keV), respectively. This explains the lower resolution, multiplicity, and poorer $R_{split}$, $CC_{1/2}$, $CC_{ano}$, $\langle I/\sigma (I)\rangle$ of the 7 keV data
[d]Values for the high resolution bin are given in the bracket
[e]
$$R_{split} = \frac{1}{\sqrt{2}} \cdot \frac{\sum_{hkl} |I_{hkl}^{even\_images} - I_{hkl}^{odd\_images}|}{\sum_{hkl} 1/2(I_{hkl}^{even\_images} + I_{hkl}^{odd\_images})}$$
[f]Denotes a negative value of $CC_{ano}$

**Table 3 Final phasing statistics**

| No. of images | Phasing Method | FOM[a] | No. residues in first round (sequenced) | No. residues in second round (sequenced) | Mean Cosine Difference[b] |
|---|---|---|---|---|---|
| 9000 | MAD | 0.529 | 127 (115) | 127 (127) | 0.372 |
|  | SAD | 0.511 | 125 (104) | 127 (127) | 0.744 |
| 6000 | MAD | 0.493 | 123 (112) | 126 (126) | 0.398 |
|  | SAD | 0.475 | 124 (95) | 124 (124) | 0.753 |
| 5000 | MAD | 0.473 | 115 (81) | 127(127) | 0.435 |
|  | SAD | 0.457 | 49 (0) | 120 (120) | 0.759 |

Comparison of SAD phasing using only 9 keV data and MAD phasing using 9 keV and 7 keV data
[a]FOM: figure of merit: cosine of the phase error as estimated by AutoSHARP[36]
[b]Reference phases were calculated from the final, refined model. The cosine difference defined as |cos[phase(final_refined_model)] − cos[phase(SHARP, obtained_with_(subset_of_images)]| was calculated to assess the quality of the phases. This is a comparison between a well-defined reference structure and the structure obtained with fewer images. By contrast, the figure of merit is an intrinsic measure without reference

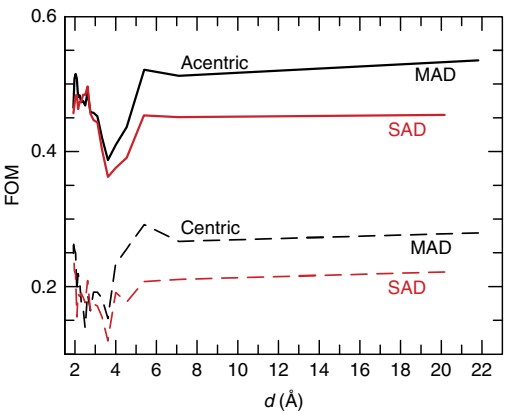

**Fig. 2** Data quality. Dependence of the final AutoSHARP[36] figure of merit before solvent flattening for centric (*dashed lines*) and acentric reflections (*solid lines*) in the SAD (*red*) and MAD (*black*) cases, using 5000 indexed images

Supplementary Fig. 10 in Supplementary Note 2). As some of the identified peaks are part of the stronger diffraction pattern, these need to be removed from the peak list for the search for peaks from the second diffraction pattern. To this end, the write_subtract.py module was implemented to remove all spots from the peak list that were closer than 10 pixels to spot positions indexed with the first colour. The remaining peak positions were then passed directly to CrystFEL's indexamajig module[35] for indexing. This procedure significantly increased the two-colour indexing rate (11.1 % (23,144 images of 208,373), see Table 1). The final structure factors for the two colours were calculated from diffraction images that were two-colour indexable (see Table 2 for data statistics and Supplementary Fig. 11).

**Phasing**. Phases were determined automatically using Auto-SHARP[36], using data to 1.9 Å resolution. This programme searches for the heavy atoms, refines their positions, B-factors and occupancies, calculates phases and performs solvent flattening. It then performs an initial round of autobuilding using BUCCA-NEER[37], followed by more solvent flattening taking the initial model into account, and then performs a final round of model building using ARP/WARP[38].

To investigate the usefulness of the two-colour phasing approach, SAD (using the 9 keV data only) and MAD (using both colours) automatic phasing was attempted with subsets of 9,000, 6,000, 5,000 and 4,000 images. At 4,000 images, both SAD and MAD failed as defined here by the failure of the programme to build the correct structure. All other attempts were successful in that >90% of the structure was built correctly in the second round of automatic building.

However, there was a clear improvement in the accuracy of the phase angle upon comparing the two-colour MAD results with the single-colour SAD results, as shown in Table 3. Plotting the estimate of the cosine of the phase angle error (figure of merit, FOM) as a function of resolution shows this improvement to mainly be seen at medium and low resolution (see Fig. 2). More importantly, at 5,000 images, the results of automatic building were clearly better in both the first and the second round of automatic building. This suggests that for difficult cases, the two-colour approach is superior.

**Discussion**

Two-colour XFEL operation[26, 39] enables new scientific applications, ranging from X-ray pump/X-ray probe experiments to the

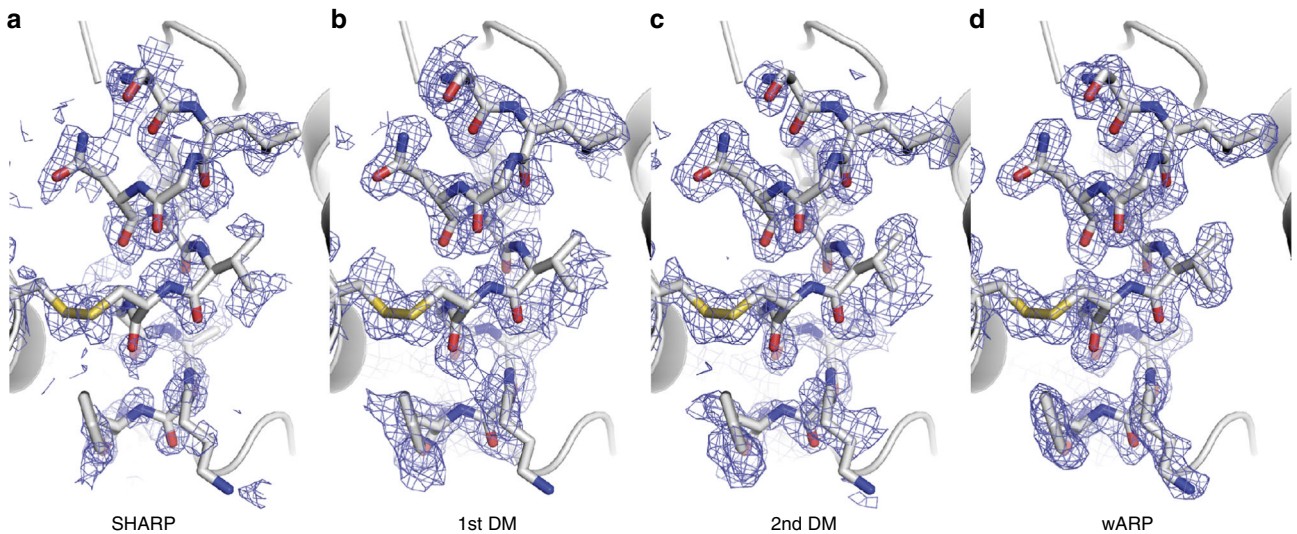

**Fig. 3** Progression of the MAD phasing process with 5,000 images. **a** SHARP[36] phases. **b** Phases after first round of density modification. **c** Phases after second round of density modification (DM, taking the first round of model building into account). **d** Phases after automatic building and -refinement by ARP/wARP[38]. All maps are contoured at 1.0 $\sigma$ and are superimposed onto the final, refined structure (PDB code 5OER)

expected use for MAD phasing of SFX data[39]. The split undulator operation at SACLA provides two-colour double X-ray pulses with large and flexible wavelength separation of more than 30%[26]. A large wavelength separation facilitates data analysis of two-colour SFX data because it ensures that most Bragg reflections in the diffraction pattern are spatially well separated and can be integrated without deconvolution, which would compromise data quality. We describe a proof-of-concept study using two-colour XFEL pulses for MAD phasing of SFX data of a lysozyme gadolinium derivative, a well-characterised model system[18, 28].

The choice of the photon energies of the two pulses depends on the energy of absorption edge(s). We chose 7 keV and 9 keV, below and above the L-edges of Gd, respectively. This yields a large anomalous signal difference and a good spatial separation of the two diffraction patterns. In fact, this photon energy difference is so large that very different regions of reciprocal space are probed. In addition to the two photon energies the ratio of their pulse energies needs to be chosen. X-ray—matter cross sections depend strongly on photon energy as can detector quantum efficiencies. The lower photon energy will give stronger Bragg intensities that are often recorded more efficiently, whereas the higher photon energy will produce much weaker Bragg intensities that are then measured with lower efficiency. The intensity ratio of the two colours can be addressed either on the machine side by changing the number of undulators used to produce each colour[26] or by inserting a filter into the X-ray beam that absorbs and thus attenuates the colour with the lower photon energy.

The analysis of two-colour SFX data is not straight-forward. In fact, direct processing with CrystFEL[35] was unsuccessful, as only a minute fraction of the hits could be indexed in both colours (see Table 1). Despite aiming for similar pulse energies for the two colours and compensating for the difference in DQE by inserting an aluminium filter, the intensity distribution of the two patterns in the diffraction image varied. One diffraction pattern typically dominated and could be indexed in one colour, but indexing of the weaker second diffraction pattern in the other colour typically failed. To index the weaker diffraction pattern, the threshold for identifying peaks had to be lowered and the previously indexed peaks were eliminated from the list. Using this approach (see Supplementary Note 2) we successfully indexed and integrated 11.1 % of the hits in both colours (see Tables 1, 2).

We deliberately used a model system with an unusually strong anomalous signal. In spite of this, we see a significant increase in the data information content of the two-colour data used for MAD phasing, as evidenced by the higher figure of merit indicating more accurate initial phases, and easier model building compared with the single-colour data SAD-phasing approach. This difference is particularly striking at 5,000 images, which is a comparatively low number for SFX data collection. Hence, these data are of lower precision than those from larger number of images, as evidenced by the data statistics (Table 2). At 5,000 images, the first round of automated building essentially failed in the SAD case, whereas in the MAD case most of the structure was built (see Fig. 3). It has been suggested that for suboptimal data, density modification might more easily improve even inaccurate phases provided by MAD, which are unimodal, rather than SAD phases which are additionally compromised by a handedness ambiguity[40]. This could help explain the superiority of the MAD phases during the later stages of structure determination. We expect the difference between SAD and MAD to be even larger for more challenging cases with weaker anomalous signals.

Traditionally, two-wavelength MAD phasing involves data collection at the peak and the inflection point (which are very close together) of an absorption edge, where the scattering properties are extremely sensitive to wavelength changes. This is challenging at XFELs owing to the inherent energy jitter of the self-amplified spontaneous emission beam. Although one could resolve such narrow energy gaps between two-colour pulses by sorting the data according to the measured per-pulse photon energy spectra after data collection, data analysis would still be extremely challenging because Bragg peaks would spatially overlap[41]. Therefore, we measured above the M- and L-edges, respectively, which, in addition, gives a large anomalous difference signal. This approach works for all elements that have more than three edges, i.e., all elements with $Z \geq 52$ (Te), which includes in particular the metals used in traditional heavy atom derivatives (Hg, Pt, Au, …). But even in the absence of a second absorption edge, the two-colour approach is likely to be highly useful for systems that are difficult to phase. When collecting SFX data with a two-colour beam that has a large energy separation, very different regions of reciprocal space are in diffraction condition simultaneously. Indexing the reflections belonging to one

colour yields the orientation matrix of the unit cell relative to the laboratory system. Future software may then use this matrix as a starting point for the initial indexing of the Bragg reflections of the second colour. Since they provide a different set of diffraction conditions, the matrix can be optimised for the second colour and through iterative refinement using the two sets of reflections, an extremely accurate orientation matrix can be obtained, in particular for the weak high resolution reflections. This is akin to the advantage of the rotation method where the initially determined orientation matrix is refined by minimising the difference in locations of predicted and observed reflections occurring in a different part of reciprocal space observed in later frames. Ideally, a global refinement including both colours should be performed but this is not possible with the currently available software. We expect that such improvements in analysis software together with detector developments increasing in particular the dynamic range will greatly facilitate two-colour data collection and MAD phasing at XFELs.

Given the emergence of and rapidly increasing demand for serial data collection at synchrotron sources[30], [42–49], this approach also requires efficient de novo phasing methods. Interestingly, it has been demonstrated previously that a dichromatic beam approach for MAD data collection is feasible at synchrotron sources[41], analogously to the experiment described here. Although the data may not be radiation damage free, it would be easy to achieve enough spatial separation between reflections, maximise the phasing signal by selecting the absorption edge or inflection point. This is feasible because of the low bandwidth at synchrotron sources, which, in addition, do not suffer from fluctuations in the relative intensity of the two colour beams.

In conclusion, we have demonstrated that XFEL-based two-colour phasing is not only feasible but also advantageous. Using a well-characterised model system we show that significantly fewer indexed patterns are required for de novo phasing using two-colour data compared with single-colour data. This should reduce the required amounts of sample and beamtime requirements. We expect two-colour data collection to be particularly useful for difficult-to-phase projects where it may make the crucial difference between being able to solve the structure and not.

## Methods

**Sample preparation and injection.** The two-colour experiment (proposal number 2015B8045) was performed in January 2016 at the Japanese XFEL SACLA in Hyogo. Lysozyme/gadoteridol microcrystals were prepared as described previously[18] except that crystal growth was done at 20 °C, resulting in larger crystals ($10 \times 10 \times 10$–$15 \, \mu m$). In brief, 2.5 ml of protein solution (32 mg ml$^{-1}$ hen egg white lysozme (Sigma) in 0.1 M sodium acetate buffer pH 3.0) and 7.5 ml precipitate solution (20 % NaCl, 6 % PEG 6,000, 0.1 M sodium acetate pH 3.0) were mixed rapidly and left over night at room temperature on a slowly rotating wheel shaker. After gravity-induced settling, the crystalline pellet was washed several times in crystal storage solution (8% NaCl, 0.1 M sodium acetate buffer, pH 4.0). At least 30 min prior to data collection, 100 mM gadoteridol (Gd$^{3+}$:10-(2-hydroxypropyl)-1,4,7,10-tetraazacyclododecane-1,4,7-triacetic acid)[28] was added to the storage solution and the crystals were left to incubate at room temperature.

A total of 7 μl of microcrystalline pellet was mixed with 75 μl grease (Super Lube) and then filled into the reservoir of a High Viscosity Extrusion injector[30]. The injector was mounted in the DAPHNIS chamber[31] which was filled with a humid helium atmosphere. Sample was extruded at a flow rate of 0.3 μl min$^{-1}$. Because of the limited dynamic range of the MPCCD detector, different crystal sizes and thicknesses of aluminium attenuators were tested in order to minimise the number of saturated reflections while keeping as much as possible of the weak high resolution diffraction.

**Wavelength determination using inline spectrometers.** A single-shot inline spectrometer was used to measure part of the Debye–Scherrer (111) diffraction rings from a diamond powder[26] using a MPCCD detector and stored as an image of $1,024 \times 512$ pixel. For the profile parameter calculation the image was collapsed into a one dimensional image of 1024 pixel; all pixel reads from the same column were summed to give rise to a double Lorentzian beam intensity profile[50].

We implemented the write_spectra.py module to perform non-linear model-fits with automatically estimated starting values for specified runs and to write the fitted parameters into a HDF5 data format file (spectra files). The energy calibration function was obtained from the comparison between the respective readings of the wide range and the narrow range inline spectrometers that were acquired during two reference runs for both photon energies (7 keV or 9 keV) (see Supplementary Note 1). The write_calib_color.py module was implemented to apply the energy calibration function to the fitted parameters obtained with the write_spectra.py module and to add the wavelength to the respective diffraction images.

**Peak identification and thresholding.** We used CrystFEL version 0.6.2. Peaks were identified by thresholding by CrystFEL's indexamajig module[35]. The initial threshold value $\tau$ was determined as the median of $\tau = \mu + 4\sigma$, the sum of the mean $\mu$ of the pixel intensity reads and its standard deviation $\sigma$ (obtained from 1,000 diffraction images). Assuming a Cauchy distribution, this corresponds to the 0.92 quantile of the pixel read values in the image. Peaks were identified using a threshold $\tau$ of 700 arbitrary detector units (ADU), and default values for the minimal signal to noise ratio (min-snr = 5) and the minimal gradient of (min-gradient = 10,000). Indexing yielded the same unit cell parameters ($a = b = 78.3$ Å, $c = 39.1$ Å, $\alpha = \beta = \gamma = 90°$) as determined previously[18]. These values were loosely imposed on the subsequent analysis steps. Deviations of the values of unit cell lengths and angles were restricted to 10% and 2%, respectively.

The final parameters were chosen such that over a wide resolution range the diffraction spots could be found by CrystFEL's indexamajig module[35]. For this purpose, the peak values and the peak background values of successfully indexed images were inspected. From the distribution of peak values above background a threshold of 200 ADU was selected. From the distribution of ratios between the peak value and the background noise a value of 5 for the signal-to-noise ratio (snr) was determined. Thus, in combination with the optional median filtering (--median-filter) the effects of the background were minimised, as the background is subtracted before thresholding takes place. From the spatial distribution of diffraction peaks (diameter 2 pixels) a mean distance of 15 pixels between two adjacent diffraction spots of a diffraction pattern was estimated. Thus a median filter with window size 16 pixels was chosen. After successful processing of the complete dataset with these values and an error analysis it was decided to increase the integration radii to (6,6,8) to compensate for the errors in diffraction spot position predictions owing to residual errors in the wavelength and detector distance estimates.

To identify the second diffraction pattern in the diffraction image, the peak search parameters were lowered to select a broader set of peaks from the image (threshold 150, min-snr 3 and min-gradient 10,000, median filter 16 pixels). The write_subtract.py module was implemented to remove all spots from the peak list that are closer than 10 pixels to peaks identified as belonging to the first diffraction pattern. The remaining peaks were indexed in the second colour.

**Data analysis and phasing.** Data analysis was performed on the SACLA High Performance Computing Cluster. For the purpose of visualisation, analysis, iteration and filtering within data processing routines written in Python, a parser for the CrystFEL[35] stream file was implemented. The stream2h5.py module scans the gigabyte-sized stream file once and transforms each line into a target data structure from which other routines (e.g., the write_subtract.py module) can extract the required information directly. The parser produces a file in HDF5 data format (which is smaller than the stream file by roughly a factor of two) to make parameters from the CrystFEL[35] stream file available in a standardised and time-efficient way.

Phasing was performed with AutoSHARP[36] Version 2.8.5, using data to 1.9 Å resolution. The 9 keV (1.38 Å wavelength) data was used either on its own for SAD phasing, or as the peak wavelength for 2-colour MAD phasing, in which case the 7 keV (1.77 Å wavelength) data were used as inflection point data. Initial estimates of f'/f'' for the 9 and 7 keV data were −4.0/11.7 e$^-$ and −10.0/3.8 e$^-$, respectively. AutoSHARP[36] searched for 2 Gd atoms using SHELXD[51], and after phasing and solvent flattening with automated optimisation of the solvent content performed two cycles of autobuilding, the first with BUCCANEER[37] and the second with ARP/wARP[38], with additional automatic solvent flattening in between. A final model refined against the 5,000 image 9 keV dataset was obtained by iterative rebuilding using COOT[52] and refinement using REFMAC5[53]. The final model displayed excellent geometry (RMSD bond lengths 0.007 Å, RMSD angles 1.6°, no Ramachandran outliers) and good R-factors (R/Rfree 0.186/0.214).

**Code availability.** Our scripts can be downloaded from https://github.com/AlexanderGorel/crystallography under the GNU General Public License v3.0.

**Data availability.** We have deposited the diffraction data reported in this study (all images collected as well as hits only) for method development in the CXIDB.org data bank with the accession code id-66 (http://cxidb.org/id-66.html). Coordinates and structure factors derived from the 5,000 images lysozyme data have been deposited in the Protein Data Bank (http://www.wwpdb.org) under the accession

code 5OER. Other data are available from the corresponding author upon reasonable request.

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

## Acknowledgements

This work was supported by the X-ray Free-Electron Laser Priority Strategy Program (Ministry of Education, Culture, Sports, Science and Technology of Japan) and partially by the Strategic Basic Research Program (JST) and RIKEN Pioneering Project Dynamic Structural Biology. We acknowledge computational support from the SACLA High Performance Computing system. The research was supported by the Max Planck Society and Dynamic Alliance for Open Innovation Bridging Human, Environment and Materials and TAGEN project of Tohoku University. We thank Dr. Roland van Gessel, Bracco Imaging Deutschland, Konstanz, Germany, for the very generous gift of the sample of gadoteridol.

## Author contributions

I.S., M.K., G.K. prepared and characterised samples, R.B.D., R.L.S., G.K., M.L.G., M.K. designed and operated sample injection hardware, M.Y., Y.J, S.I. were involved in preparations for the experiment, R.B.D., R.L.S., G.K., M.L.G., M.K., I.S., M.H., C.M.R., K.N., T.R.M.B., K.M., H.F., K.U., I.I., K.T., E.N., R.T. performed the experiment, C.M.R., M.H., K.N., T.R.M.B and L.F. performed online processing, A.G. performed off-line processing,

T.R.M.B., A.G. phased the data, L.F., T.R.M.B., I.S. jointly supervised the work, T.R.M.B. coordinated the beamtime at SACLA, I.S. designed and coordinated the project, A.G., T.R.M.B. and I.S. wrote the manuscript with input from all the authors.

## Additional information

**Competing interests:** The authors declare no competing financial interests.

