## [Peer Review file · Nature Communications]

REVIEWERS' COMMENTS:

Reviewer #1 (Remarks to the Author):

We sincerely congratulate the authors for a paper that is remarkable both in terms of novelty and elegance, but also in the writing quality. This referee acknowledges how pleasant it has been to review a quasi-flawless manuscript prepared so carefully. The idea of using two XFEL pulses of different colors to simultaneously record two diffraction patterns from the same crystals [one at the inflexion wavelength, and the other at the peak wavelength of the anomalous scatterer], and use differences between diffracted intensities to solve a structure de novo by MAD phasing is brilliant. Performing this experiment at SACLA was also appropriate, given the machine capabilities which allow separating by up to 2 keV the colors of the two simultaneous pulses. New software was developed to carry out beamline calibration and allow sequential indexing of the two lattices present in each diffraction image. Results of the authors demonstrate the feasibility of MAD phasing using two simultaneous XFEL pulses of different colors, offering new possibilities to determine structures de novo, by means of serial femtosecond crystallography (SFX). Thus, in every possible aspect, this paper represents an advancement in the field of SFX. The only possible critic to this study would be that the authors voluntarily limited themselves to a well-know, highly favorable system, i.e. the complex of gadoteridol with tetragonal lysozyme. Performing the presented experiment and data analysis on a more challenging system would have made this paper of even higher influence. We also suggest that the authors make available their scripts to the community, as these have manifestly been central to their success.

Minor comments.

Page 7:

"X-ray-matter cross sections do and detector quantum efficiencies...": suppress the 'do' ?

Page 8:

"In spite of this we see a significant increase in the data information content of the two-color data used for MAD phasing, as evidenced by the higher figure of merit indicating more accurate initial phases, and easier model building compared to the single color data SAD phasing approach" Is this a fair comparison? You obviously are using two times more data to determine the MAD phases, as compared to SAD-phases; wouldn't it be fairer to compare SAD-phasing figures of merits and model building statistics with those obtained by MAD with half the number of images ?

Page 10: Please indicate which version of crystFEL was used.

Reviewer #2 (Remarks to the Author):

In this manuscript by Gorel et al. a new method for experimental phasing using a special two color operation with large energy separation at the Japanese hard X-ray FEL SACLA is described. The authors compare the instantaneous MAD-experiment - resulting from selecting the two colors to be below and above the Gd L-edges (7 and 9 keV) - with the SAD experiment, where only the diffraction data resulting from the 9 keV pulses is used and find that the MAD experiment yields better experimental phases. The difference to a normal MAD experiment at e.g. an FEL is that since data from both energies is collected at the same time only half the sample and - at an FEL very important - only half the time of data collection is required. Thus, in combination the new method of two color MAD might overcome barriers for structure determination for samples for which no suitable search model for molecular replacement exists and which can only be produced in small quantities. The use of two color operation at an FEL for phasing is novel and since both the quantity of any interesting sample and beamtime at FELs is limited, this novel method will be of interest for others in the FEL structural biology community. Once this method has been shown to be useful for more difficult samples and weaker anomalous scatterers, it will also be of interest

for the wider field and will be helpful in advancing the application of Serial Femtosecond Crystallography to answer biological questions.

While two color operation of LCLS has been used in the past, e.g. by parts of the authors of this manuscript, to reveal the extend of radiation damage in SFX experiments, this is the first time that this mode of FEL operation has been shown to be useful beyond further understanding of X-ray matter interactions. This will influence thinking in the field and encourage others to look for other applications of special operation modes of FELs, too.

The whole work would have been a bit more convincing if a different, more complicated sample and a weaker anomalous scatterer would have been used by the authors, either instead of lysozyme-Gd or as an additional sample. However I understand that beam time at FELs is limited and that - especially - in FEL science, such experiments have been very useful and highly cited in the past. And once a proof-of-concept work has been published others will follow quickly and conduct more challenging experiments (as has happened before for both time resolved measurements and experimental phasing). Furthermore the manuscript is technically sound, well written and contain no major errors. Therefore I would suggest to publish the manuscript in Nature Communications, once minor revisions (see below) have been included.

Minor revisions:

A) The authors provide statistics on phasing and data collection. However no electron density maps and structural models have been included, nor have structural models and diffraction data been submitted to the PDB (at least the authors do not mention it). Please include a figure similar to Fig.3 in Barends et al., De novo protein crystal structure determination from X-ray free-electron laser data. Nature 505, 244 (Jan 9, 2014) and submit the data to the PDB (both at least for the case of 5000 images) or provide a good explanation why you think that this is not necessary.

B) The authors show the results for all two-color indexable images, 5000, 6000 and 9000 images. What happens between 4000 and 5000 images? What is the lowest number of images that you need for a successful MAD-experiment? How did you select the subsets? Randomly?

C) Page 1, line 5 of abstract: SFX- XFEL should be either SFX-XFEL or - better - just SFX

D) Page 1, line 7 to 9: what is „very large“? The authors show that between 6000 and 9000 indexable images are enough for successful SAD-phasing. In recent studies from SACLA (e.g. Scientific Reports 5, Article number: 14017 (2015) doi:10.1038/srep14017) only about 20000 indexable images were needed for SIRAS. At LCLS this would correspond to not more than an hour of beam time.

E) Page 2, line 4: micro crystals, should be micro- and nanocrystals, especially in regard to references 4-6.

F) Page 3, line 18: precludes precludes, should be precludes.

G) Page 7, line 16-18: isn't this true for traditional MAD as well?

Reviewer #3 (Remarks to the Author):

This paper is, as far as I am aware of, the first account of successful phasing of a protein structure de novo using

Multiwavelength Anomalous Dispersion phasing with a two colour beam at an Xfel and it should be published. The finding that the MAD phases are more accurate than the SAD phases obtained with the same number of diffraction frames, is an important result that will have an impact on phasing

experiments from microcrystals that cannot be easily carried out by conventional methods due to extreme sensitivity to radiation damage. The thorough description of data processing and the steps taken to successfully index the diffraction patterns at the weaker wavelength should help the success of future similar experiments.

Having said that, something I miss in the paper is a more detailed rationale of the choice of the energies for the experiment. Avoiding the immediate vicinity of an absorption edge seems reasonable in order to avoid extremes in the signal measured in different shots, but the region between the L3 and L2 edge would provide about the same contrast in f' , besides a higher f' , which could possibly yield better results.

I also think that it would be useful to discuss the implications of the results obtained for serial crystallography phasing experiments at synchrotrons. At the end of the fifth paragraph of the introduction, there is an isolated comment about MAD data collection at synchrotrons being done sequentially, but there is no further discussion of this point. It should be noted that the feasibility of simultaneous multicolour data collection has been in fact demonstrated at synchrotrons (eg. Kumasaka et al, Structure 2002). While sequential or interleaved data collection may offer a higher benefit-to-cost for conventional MAD experiments, right now there is a lot of interest in implementing serial crystallography facilities at intense synchrotron beamlines, so pointing out the potential advantages of a dichromatic beam approach would be very timely. Although the data would not be radiation damage free, it would be easy to achieve enough separation between diffracted spots, maximize the phasing signal by selecting the absorption edge inflection point, and there would not be fluctuations in the relative intensity of the two colour beams.

Finally, I think that the wording in the first sentence of the 5th paragraph of the discussion should be changed. "Traditionally MAD phasing is performed over a very narrow photon energy range." I believe that what is meant is that a narrow bandpass beam is desirable, to be able to maximize Bijvoet and dispersive differences by tuning to resonance and white line features of the absorption edge. However, optimal MAD phasing requires quite a wide energy range (therefore the need for a remote energy) and, as it is now written the sentence can lead to confusion.

REVIEWERS' COMMENTS:

We thank all reviewers for their very careful reading of our manuscript and their suggestions that have improved our manuscript significantly. We address the comments in detail below.

Reviewer #1 (Remarks to the Author):

We sincerely congratulate the authors for a paper that is remarkable both in terms of novelty and elegance, but also in the writing quality. This referee acknowledges how pleasant it has been to review a quasi-flawless manuscript prepared so carefully. The idea of using two XFEL pulses of different colors to simultaneously record two diffraction patterns from the same crystals [one at the inflexion wavelength, and the other at the peak wavelength of the anomalous scatterer], and use differences between diffracted intensities to solve a structure de novo by MAD phasing is brilliant. Performing this experiment at SACLA was also appropriate, given the machine capabilities which allow separating by up to 2 keV the colors of the two simultaneous pulses. New software was developed to carry out beamline calibration and allow sequential indexing of the two lattices present in each diffraction image. Results of the authors demonstrate the feasibility of MAD phasing using two simultaneous XFEL pulses of different colors, offering new possibilities to determine structures de novo, by means of serial femtosecond crystallography (SFX). Thus, in every possible aspect, this paper represents an advancement in the field of SFX. The only possible critic to this study would be that the authors voluntarily limited themselves to a well-know, highly favorable system, i.e. the complex of gadoteridol with tetragonal lysozyme. Performing the presented experiment and data analysis on a more challenging system would have made this paper of even higher influence.

We fully agree. Unfortunately, we did not have enough beamtime to do this. We are confident that we or other groups will do this in the near future.

We also suggest that the authors make available their scripts to the community, as these have manifestly been central to their success.

We agree. Therefore, our scripts can be downloaded from <https://github.com/AlexanderGorel/crystallography> under the GNU General Public License v3.0. Given the scarceness of XFEL beamtime, in particular of this unique two color mode, we have also deposited our raw data in the CXIDB.org database so that other groups can use them for algorithm development.

Minor comments.

Page 7:

“X-ray-matter cross sections do and detector quantum efficiencies...”: suppress the ‘do’ ?

We chose this wording to express that X-ray-matter cross sections always depend on photo energy whereas the dependence of detector quantum efficiencies on photon energies depends on the detector material. We rephrased to:

“X-ray—matter cross sections depend strongly on photon energy as can detector quantum efficiencies.”

Page 8:

“In spite of this we see a significant increase in the data information content of the two-color data used for MAD phasing, as evidenced by the higher figure of merit indicating more accurate initial phases, and easier model building compared to the single color data SAD phasing approach”

Is this a fair comparison? You obviously are using two times more data to determine the MAD phases, as compared to SAD-phases; wouldn't it be fairer to compare SAD-phasing figures of merits and model building statistics with those obtained by MAD with half the number of images ?

It depends what the goal of the comparison is. The referee is right that the MAD datasets constitute of significantly more data. However, we did not want to compare the information content per se but to compare how much more signal etc one gets for the same amount of sample and beamtime (i.e., “what is my money worth?”). We therefore performed the comparison using the same number of images for both cases.

Page 10: Please indicate which version of CrystFEL was used.

We used CrystFEL version 0.6.2. This is now reported in the manuscript.

Reviewer #2 (Remarks to the Author):

In this manuscript by Gorel et al. a new method for experimental phasing using a special two color operation with large energy separation at the Japanese hard X-ray FEL SACLA is described. The authors compare the instantaneous MAD-experiment - resulting from selecting the two colors to be below and above the Gd L-edges (7 and 9 keV) - with the SAD experiment, where only the diffraction data resulting from the 9 keV pulses is used and find that the MAD experiment yields better experimental phases. The difference to a normal MAD experiment at e.g. an FEL is that since data from both energies is collected at the same time only half the sample and - at an FEL very important - only half the time of data collection is required. Thus, in combination the new method of two color MAD might overcome barriers for structure determination for samples for which no suitable search model for molecular replacement exists and which can only be produced in small quantities. The use of two color operation at an FEL for phasing is novel and since both the quantity of any interesting sample and beamtime at FELs is limited, this novel method will be of interest for others in the FEL structural biology community. Once this method has been shown to be useful for more difficult samples and weaker anomalous scatterers, it will also be of interest for the wider field and will be helpful in advancing the application of Serial Femtosecond Crystallography to answer biological questions.

While two color operation of LCLS has been used in the past, e.g. by parts of the authors of this manuscript, to reveal the extent of radiation damage in SFX experiments, this is the first time that this mode of FEL operation has been shown to be useful beyond further understanding of X-ray matter interactions. This will influence thinking in the field and encourage others to look for other applications of special operation modes of FELs, too.

The whole work would have been a bit more convincing if a different, more complicated sample and a weaker anomalous scatterer would have been used by the authors, either instead of lysozyme-Gd or as an additional sample. However I understand that beam time at FELs is limited and that - especially - in FEL science, such experiments have been very useful and highly cited in the past. And once a proof-of-concept work has been published others will follow quickly and conduct more challenging experiments (as has happened before for both time resolved measurements and experimental phasing).

We fully agree with the referee that a more challenging system would have been more interesting. Unfortunately, however, we were indeed limited by the available beamtime.

Furthermore the manuscript is technically sound, well written and contain no major errors. Therefore I would suggest to publish the manuscript in Nature Communications, once minor revisions (see below) have been included.

Minor revisions:

A) The authors provide statistics on phasing and data collection. However no electron density maps and structural models have been included, nor have structural models and diffraction data been submitted to the PDB (at least the authors do not mention it). Please include a figure similar to Fig.3 in Barends et al., De novo protein crystal structure determination from X-ray free-electron laser data. Nature 505, 244 (Jan 9, 2014) and submit the data to the PDB (both at least for the case of 5000 images) or provide a good explanation why you think that this is not necessary.

We had of course thought about providing such a figure and to deposit the pdb/mtz file to the pdb. We had refrained from doing this because of the nature of the sample and the fact that we have shown previously that the lysozyme Gd system can be phased with XFEL data.

Given the comment of the referee, we have now deposited the structure and added a figure similar to Barends et al 2014 for the 5000 images case. We have also deposited our raw data so that other groups can use this very precious resource for algorithm development.

B) The authors show the results for all two-color indexable images, 5000, 6000 and 9000 images. What happens between 4000 and 5000 images? What is the lowest number of images that you need for a successful MAD-experiment? How did you select the subsets? Randomly?

The diffraction images were chosen as they were collected, i.e. hit1, hit2, hit3, ... and not randomly out of the stream. The division in subsets (4000, 5000) was random. We were not interested in the minimum number of images that allows automatic de novo phasing, it is somewhere between 4000 and 5000 images. Also previously, we did not fine tune the number of images required at the time for phasing (Barends et al Nature 2014).

C) Page 1, line 5 of abstract: SFX- XFEL should be either SFX-XFEL or - better - just SFX

We agree and changed it as suggested.

D) Page 1, line 7 to 9: what is „very large“? The authors show that between 6000 and 9000 indexable images are enough for successful SAD-phasing. In recent studies from SACLA (e.g. Scientific Reports 5, Article number: 14017 (2015) doi:10.1038/srep14017) only about 20000 indexable images were needed for SIRAS.

We removed “very” from “very large”. The referee has to bear in mind that the (anomalous) signal was very strong in the two cases mentioned (two Gd /129 residues, two Hg / 220 residues) and that the resolution was better than 2 Ang. Moreover, the native and derivative data sets of the SIRAS case were highly isomorphous. So, in this respect both cases are good model systems for method development but poor representatives of real-life, scientifically interesting and challenging systems which typically diffract to medium resolution at best, and do not have strong anomalous signals. The same applies for the examples used to demonstrate S-phasing. Again, only model systems diffracting to better than 2 Ang resolution have been phased so far. In this case at least 150,000 images were needed.

At LCLS this would correspond to not more than an hour of beam time.

This depends strongly on the properties of the system investigated (rep rate compatible with a certain sample delivery method, hit rate, indexing rate, ...).

E) Page 2, line 4: micro crystals, should be micro- and nanocrystals, especially in regard to references 4-6.

Agreed, changed.

F) Page 3, line 18: precludes precludes, should be precludes.

Changed, thank you.

G) Page 7, line 16-18: isn't this true for traditional MAD as well?

Yes. There are two differences though: 1) in traditional MAD the data are collected sequentially. Thus the weak and strong data sets are collected separately, making it easy to set peak detection parameters 2.)

Detectors used at a synchrotron for traditional MAD have a very high dynamic range. This is not true for the current generation of high frame-rate detectors used at XFELs.

Reviewer #3 (Remarks to the Author):

This paper is, as far as I am aware of, the first account of successful phasing of a protein structure de novo using Multiwavelength Anomalous Dispersion phasing with a two colour beam at an Xfel and it should be published. The finding that the MAD phases are more accurate than the SAD phases obtained with the same number of diffraction frames, is an important result that will have an impact on phasing experiments from microcrystals that cannot be easily carried out by conventional methods due to extreme sensitivity to radiation damage. The thorough description of data processing and the steps taking to successfully index the diffraction patterns at the weaker wavelength should help the success of future similar experiments.

Having said that, something I miss in the paper is a more detailed rationale of the choice of the energies for the experiment. Avoiding the immediate vicinity of an absorption edge seems reasonable in order to avoid extremes in the signal measured different shots, but the region between the L3 and L2 edge would provide about the same contrast in f' , besides a higher f' , which could possibly yield better results.

We agree. The second rationale was to have a large spatial separation between the two sets of diffraction patterns to reduce spatial overlap of reflections. We added this to the Figure legend, it is also mentioned in the text.

A good spatial separation might be more important for XFEL SFX data collection than for synchrotron multicolour rotation data collection due to the fact that only a single diffraction image is available. Given the large angular range covered in synchrotron data collection, indexing is significantly easier, also in case of overlapping reflections. It will be very interesting to see the properties of serial dichromatic beam data.

I also think that it would be useful to discuss the implications of the results obtained for serial crystallography phasing experiments at synchrotrons. At the end of the fifth paragraph of the introduction, there is an isolated comment about MAD data collection at synchrotrons being done sequentially, but there is no further discussion of this point. It should be noted that the feasibility of simultaneous multicolour data collection has been in fact demonstrated at synchrotrons (eg. Kumasaka et al, Structure 2002).

We thank the referee for pointing out this reference; we were not aware of this nice work. We rephrased the sentence to "is typically done sequentially".

While sequential or interleaved data collection may offer a higher benefit-to-cost for conventional MAD experiments, right now there is a lot of interest in implementing serial crystallography facilities at

intense synchrotron beamlines, so pointing out the potential advantages of a dichromatic beam approach would be very timely. Although the data would not be radiation damage free, it would be easy to achieve enough separation between diffracted spots, maximize the phasing signal by selecting the absorption edge inflection point, and there would not be fluctuations in the relative intensity of the two colour beams.

We agree that the two colour approach at synchrotrons is an attractive idea for serial crystallography in view of the sample consumption because of the stability of the photon energy. We added a sentence to this effect at the end of the discussion, citing Kumasaka et al as a proof of principle.

Finally, I think that the wording in the first sentence of the 5th paragraph of the discussion should be changed. "Traditionally MAD phasing is performed over a very narrow photon energy range." I believe that what is meant is that a narrow bandpass beam is desirable, to be able to maximize Bijvoet and dispersive differences by tuning to resonance and white line features of the absorption edge. However, optimal MAD phasing requires quite a wide energy range (therefore the need for a remote energy) and, as it is now written the sentence can lead to confusion.

We thank the reviewer for catching this confusing wording and changed the sentence to "Traditionally, two-wavelength MAD phasing involves data collection at the peak and the inflection point (which are very close together) of an absorption edge, where the scattering properties are extremely sensitive to wavelength changes."